# Structural Testing by Torsion of Scalable Wind Turbine Blades

**DOI:** 10.3390/polym14193937

**Published:** 2022-09-21

**Authors:** Ciprian Ionuț Morăraș, Viorel Goanță, Bogdan Istrate, Corneliu Munteanu, Gabriel Silviu Dobrescu

**Affiliations:** 1Mechanical Engineering, Mechatronics and Robotics Department, Mechanical Engineering Faculty, “Gheorghe Asachi” Technical University of Iasi, 700050 Iasi, Romania; 2Technical Sciences Academy of Romania, 26 Dacia Blvd., 030167 Bucharest, Romania; 3National Institute of Research and Development for Technical Physics, Nondestructive Testing Department, 47 D. Mangeron Blvd., 700050 Iasi, Romania

**Keywords:** wind turbine blade, structural testing, equivalent stresses, GFRP, torsion

## Abstract

In life service, the wind turbine blades are subjected to compound loading: torsion, bending, and traction, all these resulting in the occurrence of normal and tangential stresses. At some points, the equivalent stresses, due to overlapping effects provided by normal and shear stresses, can have high values, close to those for which the structure can reach to the failure point. If the effects of erosion and clashes with foreign bodies are added, the structure of the blade may lose its integrity. Considering both the complex shape of the blade and internal structure used, the mechanical behavior of the blade, such as the rigidity and resistance along the length of the blade, are usually determined with some uncertainty. This paper presents the results obtained in the non-destructive tests at static torsion of a scalable wind turbine blade. The objective of the paper was to determine the variation of the equivalent stress in the most stressed points of the blade, in relation to the torques applied. To determine the points with the highest stress, a finite element analysis was performed on the scalable wind turbine blade. Electrotensiometric transducers were mounted at different points of the blade, determining the main stresses in the respective points, as well as their variation during the torsion test, by subsequent calculations. The determinations were performed by applying the torque in both senses, in relation to the blade axis, thus concluding the values of the equivalent stress in the two cases.

## 1. Introduction

In a wind turbine, the blades are the components with the highest probability of damaging during functioning, due to impacts, loadings, or major weather phenomena. The damages of the blades can cause failure of the wind turbine and be dangerous for the vicinity [1,2,3]. If the defect can be remedied, maintenance is the next step to be taken; under real conditions, these are quite difficult to execute and, at any rate, costly. The main mechanical load applied to the wind turbine blade comes from aerodynamic air pressure [4,5]. Given the random orientation of the air currents, this stress may have different directions, with respect to the surface of the blade. In addition to aerodynamic depression, the blades are subjected to gravitational and inertial loads, with a resonance frequency that will be excited by turbulence. What is more, the sheer force of wind, as well as the existing erosive particles, lead to the emergence of additional forces and moments that increase the stresses and deformations of the blade material. Generally, wind turbine blades have a complex form and structure [6].

Usually, the blades shells are from composite materials, due their good strength to weight ratio, high stiffness, fatigue resistance, and easiness of the construction [7,8].

Considering both the complex shape of the blade and internal structure used, the mechanical behavior of the blade, such as rigidity, damping, and resistance along the length of the blade, are usually determined with some uncertainty. The calculations of distribution of rigidity, mass, and inertia along the length of the blade are achieved using experimental data [9].

In order to maintain the reliability of operation, the wind turbine blades must be tested, according to international standards, in order to further obtain IEC, GL, and DNV certifications [10].

IEC 61400-23:2014(EN) standards defines the requirements for the structural testing of wind turbine blades, as well as for the interpretation and evaluation of the results obtained. The following tests are considered: static loads, fatigue, static loads, and after fatigue tests, as well as tests that determine other blade properties. The purpose of the tests is to demonstrate that the blade will sustain, without failure, all the exploitation loads, including the accidental ones. In addition, we can try to optimize the shapes and dimensions of the blade, so that the resulting stresses are distributed as evenly as possible, taking a reduction in the total weight of the blade into account. It should be borne in mind that the blade, being the most requested component of the whole wind turbine assembly, should not deteriorate before its end of life. The wind turbine blades are subjected to intense cyclical stresses throughout their lifetime. Deterioration of a structural element, as a result of repeated charging cycles or environmental action, is generally represented in crack, fatigue, or impact. The lifetime represents the number of charging cycles or equivalent time before the first signs of damage are observed. The lifetime depends on factors, among which, the most important are: the variability and load history, manufacturing method, materials and structure used, and size of blade [11,12,13] Moreover, the anisotropic characteristics of the composite material used can lead to results that are difficult to define when determining the presented stresses when using Finite Element Analysis, FEA [14,15,16].

The testing of blades can be carried out at full scale, both static and dynamic, taking all the damages that can possibly appear, due impacts, environmental erosion, etc., into account [17]. The loading test at full-scale should satisfy the IEC 61400-23-2014 (EN) requirements.

During the static tests of wind turbine blades, ultimate forces (stresses) and the displacement (strains) can be determined. Additionally, the shape, dimensions, structure, and material properties can be confirmed.

At static tests, the loading can be extended to 150% relative to nominal load. In order to obtain results to failure, the test equipment must have high precision. The stiffness and strength tests should validate and certificate the FEA simulations. During static tests, the forces can be applied by hydraulic actuators, with the displacements being in relation with variation of loading forces in time Figure 1, [18], by weights, or mixed variants; in all cases, the forces and displacements should be measured.

In experimental research [19,20], static bending tests were performed, and it was shown that the blade can withstand the extreme conditions predicted for the design and sensors used, as well as provide valuable information on the structural properties of the blade, i.e., the displacements, deformations, and stresses.

Torsional testing [21,22] aimed to study the failure of the weakest joint, the one between the blade in the composite material and aluminum alloy piece in the area of the clamping hub, created by adhesive bonding of epoxy adhesive. Following loading, failure occurred in the hub-body joint area. Depending on the size of the wind turbine blades, it can end up being that the natural torsional frequency overlaps some of the bending modes. In this case, catastrophic failure due to flutter instability can occur. It is very important to be able to make an acceptable prediction of the torsional behavior of the blade superimposed on the bending loads. In this respect, static torsional tests have been performed in [23], which, together with finite element analyses, provide conclusive results on the torsional modes. To determine the effect of blade vibration on aerodynamic performance, high-resolution numerical simulations of an oscillating wind turbine blade were performed in [24,25]. It is found that pressure fluctuations are amplified by the oscillation, thus leading to a greater variation in fatigue loading.

Other static loading of the blade is generally carried out by means of crank-driven screws, electric winches, or hydraulic actuators. For small shovels, loading can be achieved with weights, such as sandbags or ingots, metal silt sacks placed on the blade, or hanging water-filled pots on the blade. However, there is a risk the blade may break because the load cannot be quickly removed when the blade starts to give way [26,27].

## 2. Materials and Methods

### 2.1. Materials

The material used in the manufacture of the blade and specimens is RT 500 glass fiber reinforced composite (GFRP) and EPIKOTE epoxy resin. Three test specimens were cut from the composite plate, in three directions (transverse (TR), longitudinal (LG), and at 45°) (Figure 2a), composed of 10 layers oriented at (0°/90°) (Figure 2b).

### 2.2. Tensile Tests

Three samples were tensile tested, with specimen sizes conforming to ASTM D3039. The tests were carried out on the INSTRON 8801 universal machine that developed a maximum force of 100 KN. The test regime was set by increasing the extension rate to 0.5 mm/min. Due to the constant speed of movement during the test, we have the possibility to observe the occurring deformations and a good accuracy of the drawing of the characteristic curve. The stresses were held until breakage, as can be seen in Figure 3.

### 2.3. Electron Microscopy Analyses (SEM)

Electron microscopy analyses (SEM) used samples of the GFRP composite on the plate directions of the experimental material and another sample of the base material. The samples were coated with a 5 nm Au layer using the Luxor Au SEM COATER, CT-2201-0144 (Kusterdingen, Germany), in order to increase the electrical conductivity of the samples. Surface morphological determinations were performed with the SEM FEI Quanta 200 3D microscope (Brno, Czech Republic), using the following parameters: low vacuum mode, LFD (large field detector), HV (high voltage): 20 kV, WD (working distance): 10 mm, and spot size: 5.

### 2.4. Test Device Description and Finite Element Analysis

The current status of the achievements in the monitoring, diagnosis, and systematic laboratory testing of turbine blades made of GFRP at national level is not very extensive. In this respect, a torsion test stand has been designed and built in Figure 4 and Figure 5. In order to achieve only torque load, the blade was fixed at one end, with the other end being only allowed to rotate the Z-axis (geometric axis of the blade). At the free end of the blade, a sledge device and centering tip (Morse taper) were used to prevent the translation of this end from any axis, but with the possibility of rotating the blade along in longitudinal axis. In this way, by attaching the weights to the load lever, the blade was load only at torsion, thus eliminating the possibility of bending. The torsion moment is inserted by the lever by attaching the weight one end. At the same time, the angle of rotation of the lever was measured with an angular device.

A scalable E-glass composite blade structure and reinforcement system was designed. The dimensions of the wind turbine blade (WTB) have been analyzed, and the blade and rigging (reinforcing system), as well as the attached devices (blade matrix and moldings for tensile and torsional stresses) have been designed. The blade has an overall length of 1750 mm, maximum chord of 273.71 mm, chord tip at 81.54 mm, an angle of entry of 55.62°, and angle of attack 29.74°, with full pitch control to reduce aerodynamic cyclic loads. The blade attachment for the rotor is via a 100 mm diameter coupling. The blade was designed with a right leading edge and right taper trailing edge. A NACA airfoil profile was used for structural performance. In order to achieve increased structural strength and stiffness at 0.286R (R—total blade radius—distance from the center of the rotor to the tip of the blade), the same NACA profile was applied to both the upper and lower surfaces. The configuration of the wind turbine blade (WTB), type GE1.5sle, was determined by studying the parameters using the finite element method (FEM). The FEM analysis proved that the joint is safe under twist loading conditions. The determination of the critical zones of the WTB, scalable during load simulation using FEA, shows that the constituent elements of the respective airfoil I profile respects the same critical zones [28].

The blade was considered fixed in the hub, and a maximum torque (torsion moment) of 60 Nm was applied at its top (Figure 6). The torsion moment was oriented along of the Oz axis, and the direction of rotation was in the trigonometric (market with 2 at test) and clock-wise (market with 1 at test) directions.

From the discretization step, a total of 1,338,842 nodes and 832,563 elements were determined. Elements in FEA are generally grouped into 1D, 2D, and 3D elements. They are recognised according to their shapes. In the present case, we used 3D solid tetrahedral elements. The maximum element size was 8 mm for the turbine blade, and the Sizing function was used to increase the number of elements in the required areas: on the trailing edge: 0.5 and 1 mm elements, and in the transition area from the cylindrical part of the blade to the blade profile: 3 mm elements. For the hub in the embedment area and on the rigid structure, we used 5 mm elements (Figure 7).

Blade used to determine the maximum stresses introduced as a result of torsional stress was a 1:10 scale model. In order to determine the location of the strain gauge transducer, a finite element analysis of the blade model was made using the ANSYS Academic 17.2 program. In the simulation, two types of material were used, from which the three components of the assembly were:Structural steel (blade hub);Epoxy E-Glass UD (for the blade and the central stiffening element).

The FEA determined the stresses (Figure 8) and displacements in the blade. The maximum von Misses stresses was 139.66 MPa, registered at the tip of the blade.

Taking into account that the torque remains constant along the entire length of the blade, it is noted that von Misses stresses are kept approximately constant on the tip of the blade, and they are smaller in the fixed area. However, in the area with strong section change, there is an increase in stresses, as determined by the phenomenon of concentration of stresses.

On the blade four electrotensiometric transducers were mounted, type Vishay C2A-06-062WW, with attached wires, each having three strain gauges, with 350 Ω electrical resistance (Figure 9).

For the application of these transducers, the blade surface was machined by removing the filler layer applied on the composite structure (glass fiber reinforced polymer) with P100 (401) sandpaper. After removal of the resulting powder, the surface was wiped with a cotton cloth soaked in ethyl alcohol (STF 247/2002). The rosettes were bonded using HBM’s Z-70 (cyan-acrylate) adhesive, the polymerization of which was accelerated with Vishay’s 200 catalyst-C. The orientation of the rosettes sought to measure the effects of the blade twist using marks oriented at 90° to each other (marks a and c in Figure 9); therefore, mark b of the transducers was oriented parallel to the blade axis.

By orienting the transducer, it has been sought to measure the effects of twisting by means of the strain gauge oriented 90° to one another. The transducers were applied in two sections: at 230 mm of the clamping end, beyond the clamping area (transducers T1 and T2), as well as 130 mm from the blade tip (T3, T4) (Figure 10). T1 measures the deformations of the blade in the plane of the reinforcement profile (variable height I profile), and T2 measures the deformations of the blade wall in the hollow area, laterally to the reinforcement profile. The transducers T3 and T4 measure deformations in hollow areas adjacent to the reinforcement profile.

In Figure 11, the overview image of the blade is presented, as well as the location of the four transducers and details of the location on the strain gauges in the considered sections.

Each strain gauge was connected to a channel of a Vishay P3 tensor bridge, and the type of mounting was a quarter bridge, with the balancing of the marks being performed automatically in the P3 bridge. As there were four transducers, with three strain gauges each, twelve acquisition channels were required, which is why three tensor bridges, each with four channels, were used.

## 3. Results

Figure 12 and Figure 13 represent the SEM surface and cross-section images of the GFRP base material, as well as the tensile specimens for the three directions samples. Figure 12A highlights the morphological aspect of the surface of the base material, having a compact, homogeneous surface, without microcracks, but with small porosities. As shown in Figure 12B, the 10 polymer layers in the GFRP cross-section, which is part of the chemical composition. These microstructures are in agreement with the studies performed by [31].

Figure 13 shows the cross-section images for the three directions of stress. Thus, Figure 13A highlights the fiber breakage in the longitudinal direction, and Figure 13B shows the fiber breakage at 45°; these two breaks having a primary damage aspect. In the case of Figure 13C, delamination at the intermediate layers were observed, and the breakage had a medium behavior.

Figure 14, Figure 15 and Figure 16 illustrate the surface appearance of the tested specimens for all three directions of stress. In all three cases, a similar breakage mechanism was observed in the fiber–resin matrix, showing a brittle-type breakage. The different image magnitudes show the fracture behavior of the polymer pull-out fibers at the resin interface, as well as their appearance after breakage. A more compact and evenly distributed morphology of these fibers is observed in Figure 14, while Figure 15 and Figure 16 show a more pronounced dislocation.

Using the device of Figure 4, the torsion moment was applied in both senses to the experimental determinations via the mounting of the lever, once in the position indicated in Figure 4; the second time, the lever was rotated by 180, relative to this position. As mentioned before, four electrotensiometric transducers were glued to the blade, each of which had three strain gauges. As a result, for each load direction, 12 strains were acquired at each load stage. In Table 1, the load blade data is presented at different loading steps. Thus, the initial loading was performed only by means of the load arm, visible in Figure 4, with a mass of 9.4 kg, and the distance from its center of gravity to the joint at the end of the blade was 644 mm. As a result, this load created a torsion moment of 59,385.82 N·mm. Each time, a new weight was added to the previous load. The four-loading stages, illustrated in Table 1, were kept for both directions. At the start of the load, the lever was locked against rotation and initial loading.

At the same time, for each application step, the angle of rotation of the load arm was measured for each torsion moment.

Based on the acquired data, a number of specific characteristics have been determined; their significance and calculation formulas are presented below. With the calculation formulas shown below, there were determined: the maximum normal main stress, *σ_max_*, minimum normal main stress, *σ_min_*, maximum shear stress, τ_max_, and angle of the direction of the maximum main normal stress, direction *θ*.

The calculation formulas for these characteristics is outlined below [26].

The maxim normal main stress, *σ_max_*:(1)$$\sigma_{max}=\frac{E}{2\left(1-\nu^{2}\right)}\left[\left(1+\nu \right)(\varepsilon_{a}+\varepsilon_{c})+\left(1-\nu \right)\sqrt{2\left[{\left(\varepsilon_{a}-\varepsilon_{b}\right)}^{2}+{\left(\varepsilon_{b}-\varepsilon_{c}\right)}^{2}\right]}\right]$$The minimum normal main stress, *σ_min_*:(2)$$\sigma_{min}=\frac{E}{2\left(1-\nu^{2}\right)}\left[\left(1+\nu \right)(\varepsilon_{a}+\varepsilon_{c})-\left(1-\nu \right)\sqrt{2\left[{\left(\varepsilon_{a}-\varepsilon_{b}\right)}^{2}+{\left(\varepsilon_{b}-\varepsilon_{c}\right)}^{2}\right]}\right]$$The maximum shear stress, *τ_max_*:(3)$$\tau_{max}=\frac{E}{2\left(1+\nu \right)}\left[\sqrt{2\left[{\left(\varepsilon_{a}-\varepsilon_{b}\right)}^{2}+{\left(\varepsilon_{b}-\varepsilon_{c}\right)}^{2}\right]}\right]$$The angle of the direction of the maximum main normal stress, direction *θ*:(4)$$\theta =\frac{1}{2}arctg\left[\frac{2\varepsilon_{b}-\varepsilon_{a}-\varepsilon_{c}}{\varepsilon_{a}-\varepsilon_{c}}\right]$$
where *ε_a_*, *ε_b_*, and *ε_c_* are deformations measured with the three strain gauge marks of the rosette in Figure 9b.

To determine the angle of the direction of the maximum main normal stress, θ, the following are achieved.

The value of the angle determined by the Formula (4) is in radians and must be converted to degrees.Depending on the sign of numerator and denominator of the fraction in relation (4), the final angle of the maximum main deformation direction is calculated with one of the following relations:
If numerator ≥ 0 and denominator > 0 then φ = |θ|.If numerator > 0 and denominator ≤ 0 then φ = 90° − |θ|.If numerator ≤ 0 and denominator < 0 then φ = 90° + |θ|.If numerator < 0 and denominator ≥ 0 then φ = 180° − |θ|.


Figure 17, Figure 18, Figure 19 and Figure 20 show the variation of the maximum normal main stress *σ_max_*, minimum normal main stress *σ_min_*, and maximum shear stress *τ_max_*, as reported to the angle of the rotation of the torque end for all eight cases, respectively, for the four transducers placed in the two methods of loading. Young’s modulus and Poisson’s coefficient were determined on the basis of the tensile tests performed in previous steps [32]. The values of these characteristics were: E = 36,233 MPa, respectively, ν = 0.161. As a result, these values were used in the above-mentioned computing formula.

In the finite element analysis (Figure 8), it was found that the highest value of the von Mises stress was obtained at the tip of the blade, corresponding to the location of the T4 transducer (Figure 11). Considering the von Mises relation: $\sigma_{vM}=\sqrt{\sigma_{max}^{2}+3\tau_{max}^{2}}$, from the experiment, we deduce that:(5)$$\sigma_{vM\;T4}=\sqrt{80^{2}+3\times 60^{2}=131.14\;\mathrm{MPa}}$$
where the stresses calculated from the strains provided by the T4 transducer at a torque of 59.38 N·m were approximately: *σ_max_* = 80 MPa and *τ_max_* = 60 MPa. It was found that close values of the von Mises stresses from the FEA and experiment were obtained, respectively: AEF: torque: 60 N·m, *σ_vM_* = 139.66 MPa; test: torque: 59.38 N-m, *σ_vM_* = 131.14 MPa.

From the static tensile loading of a specimen, made of same material as the blade and drawing of the characteristic diagram, a creep stress of about 430 MPa was obtained, so that the maximum stress observed, at any of the four transducers, was lower than the yield stresses.

As mentioned above, the loading was performed in both directions. It has been found that there are differences in the angle of the rotation of the loading arm for the two loading directions. It is noted that the angle of rotation of the loading arm is equal to the angle of rotation of the free end of the blade. Thus, Figure 21 shows the variation of angle of rotation of the boom, with the moment of torsion applied to the blade for the two load senses.

It can be seen that the values of the angle of rotation are higher for the trigonometric loading sense. In Figure 4, the load is presented in the clockwise direction.

## 4. Conclusions

Studying Figure 17, Figure 18, Figure 19 and Figure 20, the following observations can be made:The variation in maximum and minimum stresses, with regard to the angle of rotation of the unembedded blade end, were generally linear. Thus, it can be concluded that the test was made in the elastic field.The highest stresses were obtained in the area where transducer 4 was mounted (Figure 9).It has been proven by the results of the Finite Element Analyses.The highest value of the maximum main normal stresses, experimentally determined with transducer 4, in rotating direction 1 (clockwise direction), was 190 MPa.The lowest value of the minimum main normal stresses experimentally determined with the transducer 4 in rotating direction 2 was −187 MPa.Finite element analysis resulted in a maximum normal tension of about 140 MPa.

The different from the experimental values can be explained by the fact that the FEA was considered an ideal composite material, which was perfectly balanced with all layers of fiberglass. However, for the realization of the blade model, we also used chopped fiber layers, but only in the area of the attack board. For a perfectly balanced composite material, the torque-angle rotation diagram should not depend on the direction of rotation; however, experimentally, there are some differences that depend on the direction of rotation.

The values of the angle of rotation of loading arm (the free end of the blade) were higher for the trigonometric loading direction. This is explained by the geometric asymmetry of the blade.

The blade torsional tests presented in this paper did not endanger the integrity of the blade, as a result of the FEA.

The differences between experimental and FEA values are acceptable, thus demonstrating that the numerical model is valid.

The turbine blade subjected to the experiment had a ratio of 1:10 to the real in-situ blade. By loading the turbine blade in the elastic range, at the maximum moment of 170 Nm, maximum stresses lower than the yield stress were obtained. Under these conditions, it can be said that the blade, in real dimensions, can withstand at torques up to 1700 Nm, with a safety factor of c = 230/190 = 1.2.

## Figures and Tables

**Figure 1 polymers-14-03937-f001:**
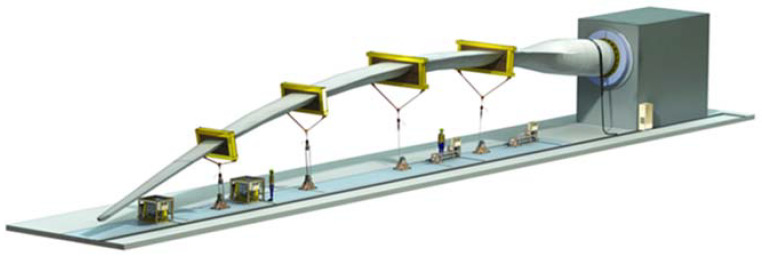
Mode of application of forces to static bending. Reprinted with permission from Ref. [18] Copyright 2022, MTS Wind Turbine Blade Static Test Solutions, Available online: https://www.mts.com/en/applications/energy/wind-turbine-testing (Accessed 6 August 2022).

**Figure 2 polymers-14-03937-f002:**
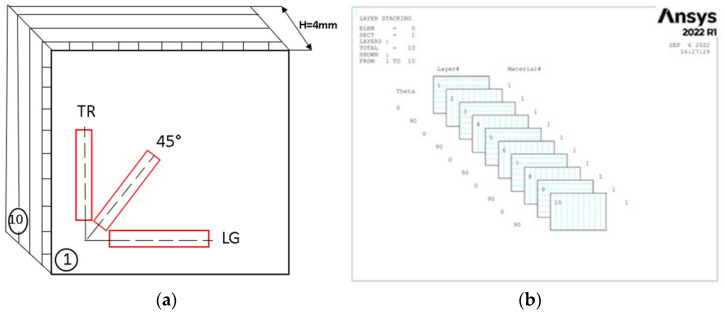
(**a**) Directions for sampling from the GFRP plate (emphasized with red frame) (**b**) and configuration of layers composite (0°/90°).

**Figure 3 polymers-14-03937-f003:**
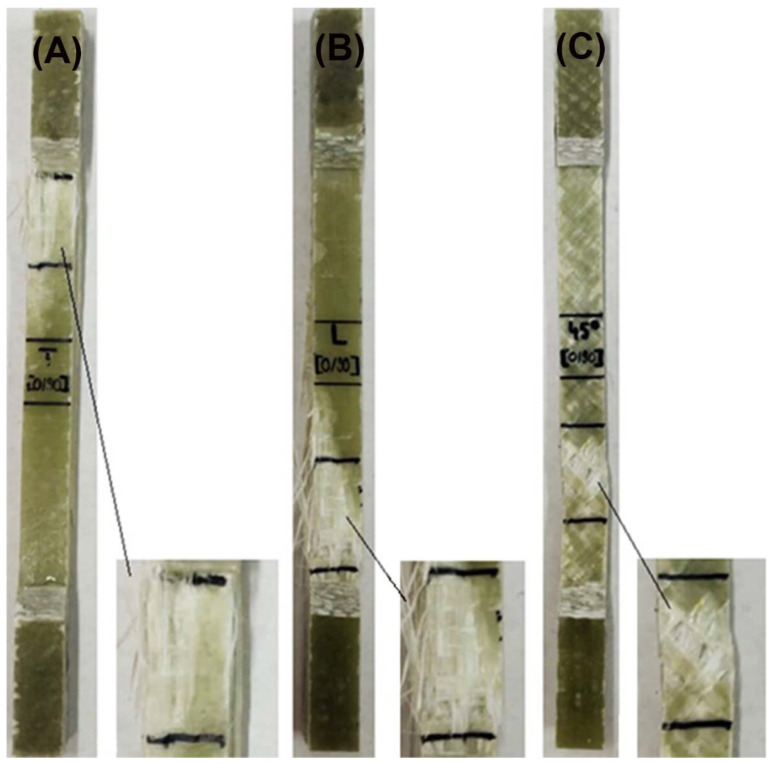
Typical failure modes of GFRP laminates: (**A**) transversal, (**B**) longitudinal, and (**C**) oriented at 45°.

**Figure 4 polymers-14-03937-f004:**
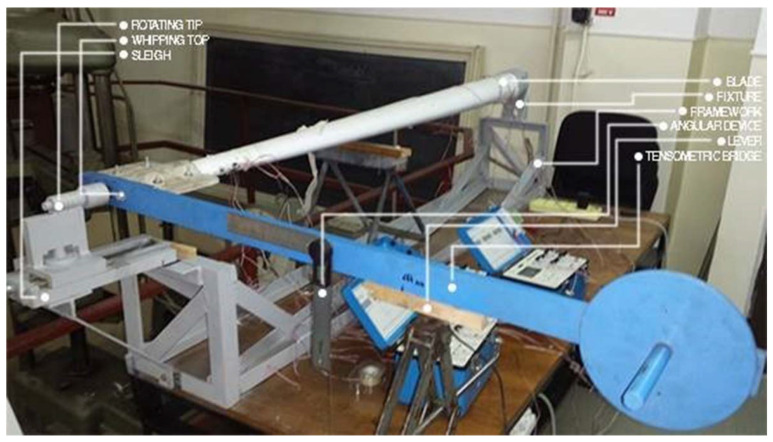
Torsion device with mounted blade model.

**Figure 5 polymers-14-03937-f005:**
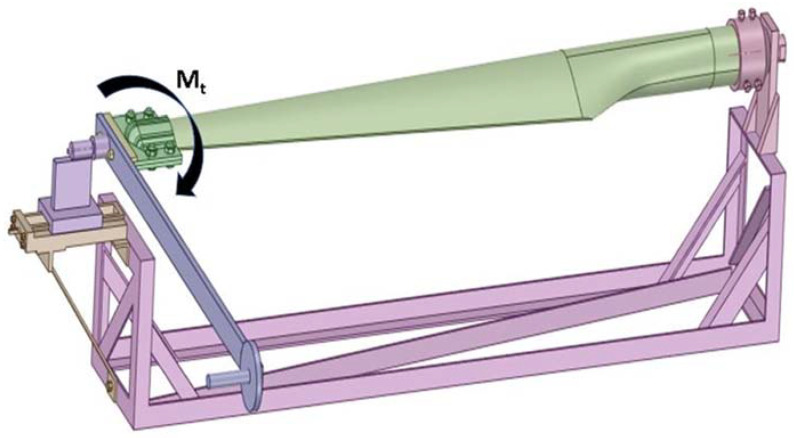
The 3D model of the device for blade torsion test.

**Figure 6 polymers-14-03937-f006:**
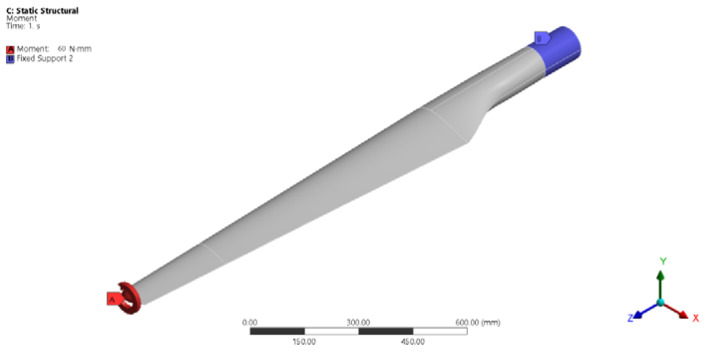
Support and loading scheme.

**Figure 7 polymers-14-03937-f007:**
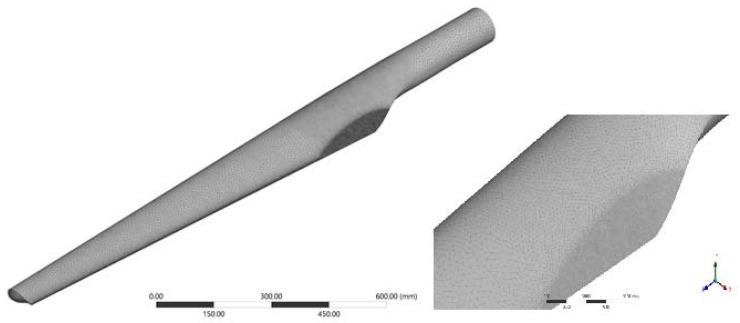
Discretization of the model.

**Figure 8 polymers-14-03937-f008:**
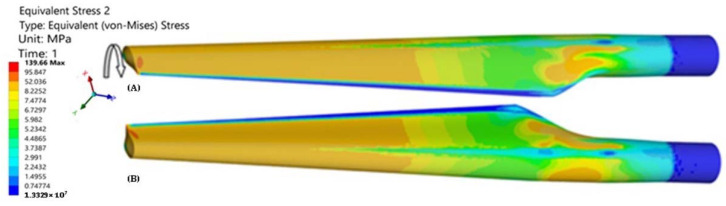
The gradient of stresses in the blade: (**A**) extrados; (**B**) intrados.

**Figure 9 polymers-14-03937-f009:**
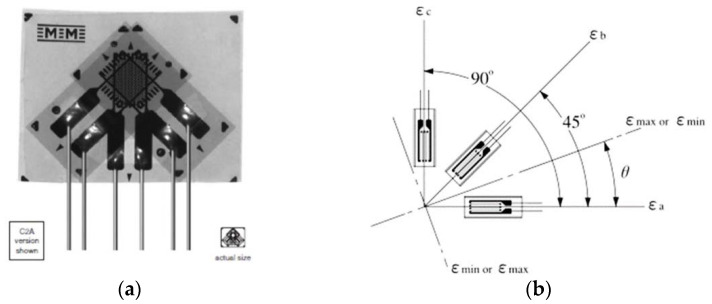
The transducer (rosette) of three superimposed marks, with the axes offset by 45° [Reprinted with permission from Ref. [29] Copyright 2022, General Purpose Strain Gages-Stacked Rosette. C2A-06-062WW. Available online: https://docs.micro-measurements.com/?id=2560 (Accessed 5 July 2022)], used in the present experiment (**a**); notations used in the work (**b**) [Reprinted with permission from Ref. [30] Copyright 2022, Method of Obtaining Magnitude and Direction of Principal Stress (Rosette Analysis). Available online: https://www.kyowa-ei.com/eng/technical/notes/technical_note/rosette_analysis.html (Accessed 2 August 2022)].

**Figure 10 polymers-14-03937-f010:**
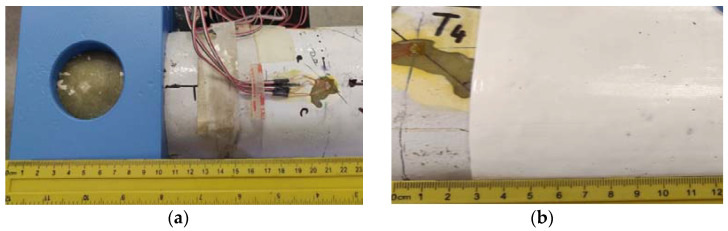
Measuring section located 230 mm from grip end (**a**) and 130 mm from blade tip (**b**).

**Figure 11 polymers-14-03937-f011:**
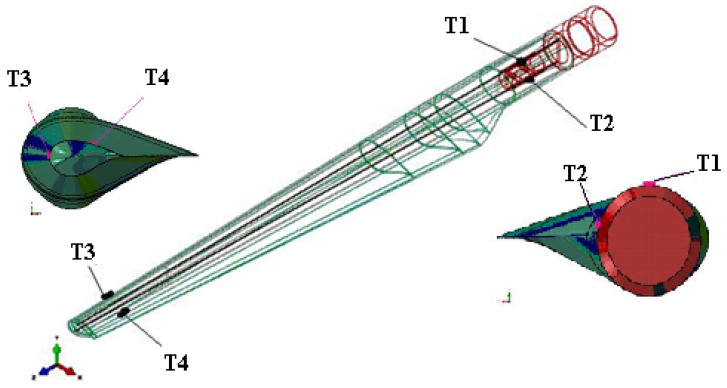
Overview and details of the location of the transducers.

**Figure 12 polymers-14-03937-f012:**
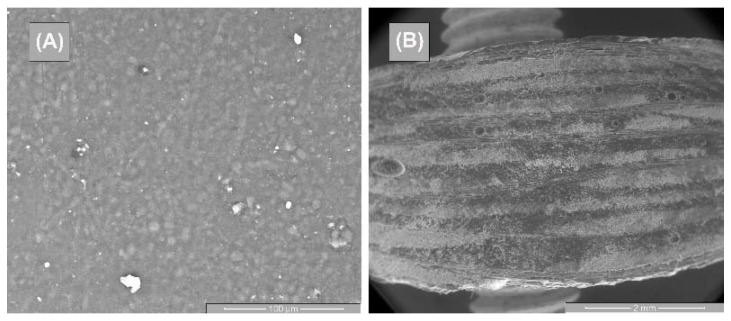
SEM images of the base material—GFRP: (**A**) surface image and (**B**) cross-section image.

**Figure 13 polymers-14-03937-f013:**
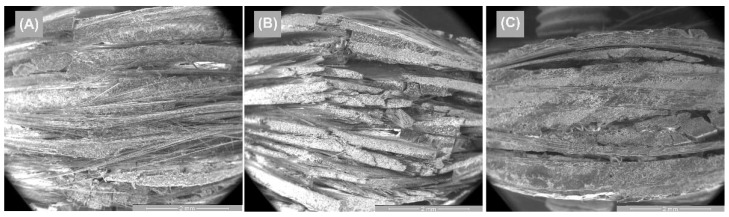
Cross-section SEM images of GFRP specimens in the 3 stress directions: (**A**) transversal, (**B**) oriented at 45°, and (**C**) longitudinal.

**Figure 14 polymers-14-03937-f014:**
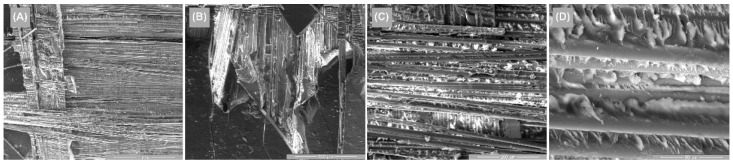
Surface SEM images of transversal 0–90 GFRP sample: (**A**) 100×, (**B**) 200×, (**C**) 500×, and (**D**) 2000×.

**Figure 15 polymers-14-03937-f015:**
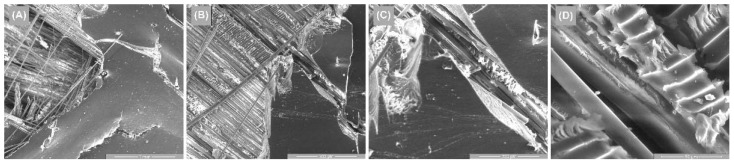
Surface SEM images of oriented at 45 GFRP sample: (**A**) 100×, (**B**) 200×, (**C**) 500×, and (**D**) 2000×.

**Figure 16 polymers-14-03937-f016:**
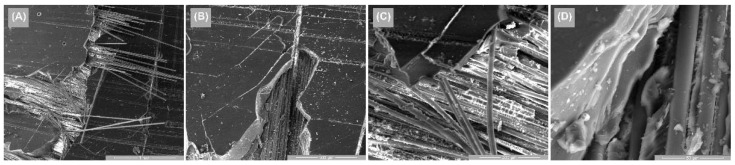
Surface SEM images of longitudinal 0–90 GFRP sample: (**A**) 100×, (**B**) 200×, (**C**) 500×, and (**D**) 2000×.

**Figure 17 polymers-14-03937-f017:**
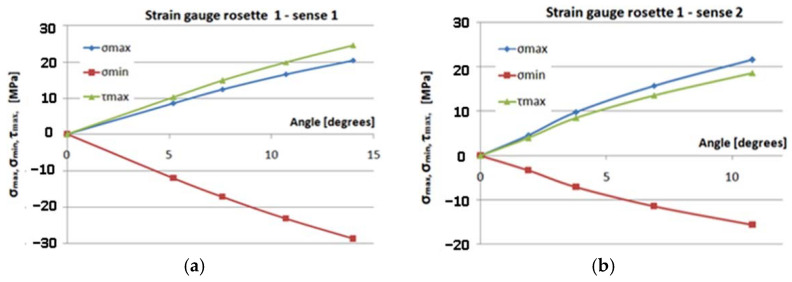
The variation of the maximum normal main stress *σ_max_*, *σ_min_*, and *τ_max_*, in relation to the angle of rotation. (**a**) Sense 1; (**b**) sense 2.

**Figure 18 polymers-14-03937-f018:**
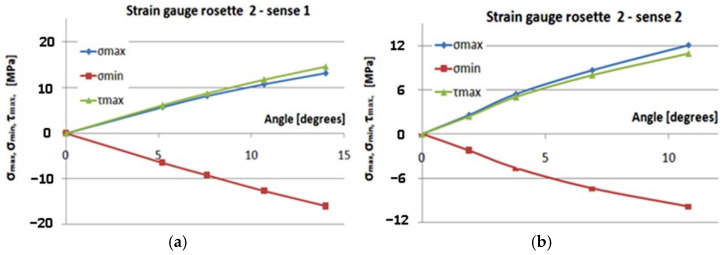
Gauge2–sense 1,2: the variation of the maximum normal main stress *σ_max_*, *σ_min_*, and *τ_max_*, in relation to the angle of rotation. (**a**) Sense 1; (**b**) sense 2.

**Figure 19 polymers-14-03937-f019:**
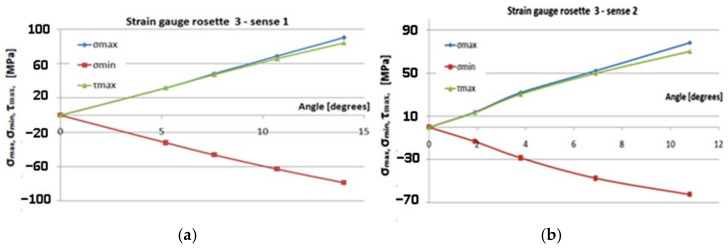
Gauge 3–sense 1,2: the variation of the maximum normal main stress *σ_max_*, *σ_min_*, and *τ_max_*, in relation to the angle of rotation. (**a**) Sense 1; (**b**) sense 2.

**Figure 20 polymers-14-03937-f020:**
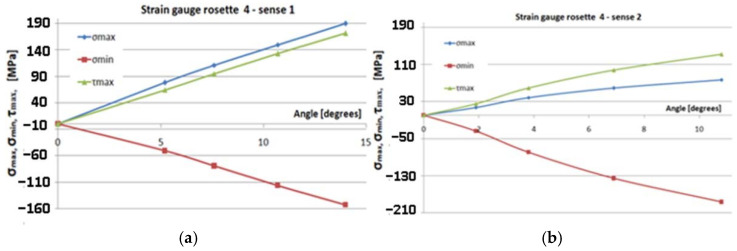
Gauge 4–sense 1,2: the variation of the maximum normal main stress *σ_max_*, *σ_min_*, and *τ_max_*, in relation to the angle of rotation. (**a**) Sense 1; (**b**) sense 2.

**Figure 21 polymers-14-03937-f021:**
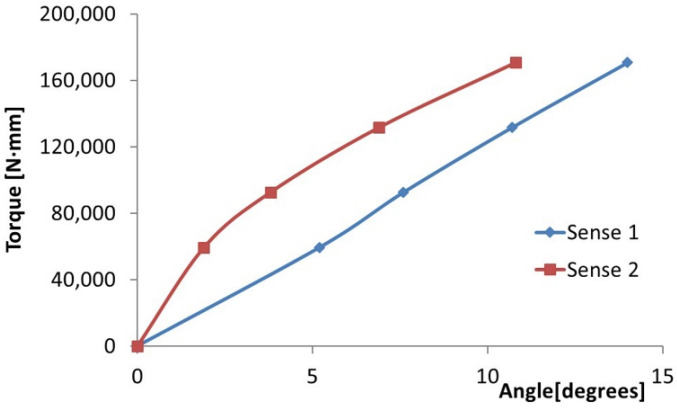
Variation of the angle of rotation of the arm with the torque moment.

**Table 1 polymers-14-03937-t001:** Load data and measured effect (bending angle of the blade).

			Sense 1	Sense 2
Arm [mm]	Weight [kg]	Torque [N∙mm]	Angle (°)	Angle (°)
-	0	0	0	0
644	9.4	59,385.82	5.2	1.9
995	+3.4	92,573.05	7.6	3.8
995	+7.4	131,616.8	10.7	6.9
995	+11.4	170,660.6	14	10.8

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
