# Peer review of "Structural Testing by Torsion of Scalable Wind Turbine Blades"

_polymers, 2022, doi:10.3390/polym14193937_

Round 1

Reviewer 1 Report

1.       More physical discussion about the impact of turbine blade torsion on their performance is necessary.

2.       The introduction can be improved. For wind turbine blade torsion and bending, the authors can refer to doi.org/10.1016/j.renene.2022.04.052. For high resolution simulations of wind turbine blades: doi.org/10.1016/j.energy.2021.120261.

3.       Figure captions are too long and must be shortened. Captions of figures 9-12 are 12 lines.

Author Response

Responses Reviewer 1.

Dear reviewer,

Thank you very much for your positive feedback. You ask to clarify more details on the GFRP materials and the novelty of our work, which certainly would give an improvement for our research. So, thank you for these valuable comments. We have tried to answer all aspects that were mentioned.

  1. More physical discussion about the impact of turbine blade torsion on their performance is necessary.

Response 1.

In the paper, in Chapter 1, lines 95-99, some elements and a bibliography on the impact of torsion on the structural integrity of the blade were introduced. These are:

Depending on the size of the wind turbine blades, it can end up that the natural torsional frequency overlaps some of the bending modes. In this case, catastrophic failure due to flutter instability can occur. It is very important to be able to make an acceptable prediction of the torsional behavior of the blade, superimposed on the bending loads. In this respect, static torsional tests have been performed in [23] which, together with finite element analyses, provide conclusive results on torsional modes.”

  1. The introduction can be improved. For wind turbine blade torsion and bending, the authors can refer to doi.org/10.1016/j.renene.2022.04.052. For high resolution simulations of wind turbine blades: doi.org/10.1016/j.energy.2021.120261.

Response 2.

In the introductory chapter, we introduced references to the behavior of turbine blades in bending and torsion, and to high-resolution simulations of wind turbine blades lines 100-104

To determine the effect of blade vibration on aerodynamic performance, high-resolution numerical simulations of an oscillating wind turbine blade were performed in [24, 25]. It is found that pressure fluctuations are amplified by the oscillation, leading to a greater variation in fatigue loading.”

  1. Figure captions are too long and must be shortened. Captions of figures 9-12 are 12 lines.

Response 3

On page 11, lines 325 to 328 the authors have modified the legends of figures 16-19 have been reduced in size.

We hope that we have answered all the topics you mentioned and also, we have added to the paper some material analysis in order to highlight the results of the Journals topics.

Reviewer 2 Report

The research was carefully and adequately framed and is presented in a transparent and comprehensive way. Additionally, the paper presents a set of interesting and useful results but, in my opinion, is out of interesting for the audience of Polymers (Advanced Materials and Polymers for Industrial and Biomedical Applications). I should encourage authors to submit their work to a journal dedicated to the topic, in order to gain notoriety in the scientific community related with the numerical and experimental study of structural behaviour of composite materials.

Author Response

Reviewer 2

Dear reviewer,

Thank you very much for your positive feedback. You ask to clarify the interest of the audience of Polymers (Advanced Materials and Polymers for Industrial and Biomedical Applications),  more details on the GFRP materials, and the novelty of our work, which certainly would give an improvement for our research. So, thank you for these valuable comments. We have tried to answer all aspects that were mentioned.

The research was carefully and adequately framed and is presented in a transparent and comprehensive way. Additionally, the paper presents a set of interesting and useful results but, in my opinion, is out of interest for the audience of Polymers (Advanced Materials and Polymers for Industrial and Biomedical Applications). I should encourage authors to submit their work to a journal dedicated to the topic, in order to gain notoriety in the scientific community related to the numerical and experimental study of the structural behavior of composite materials.

Response 1

In order to demonstrate the novelty of the research and the approach to the topic of the Special Issue ” Advanced Materials and Polymers for Industrial”, we considered it necessary to introduce as a separate chapter some SEM morphological analyses on these GFRP materials (from which the wind turbine blade was made), both in the base material and the required tested samples. In this way, please accept that these studies contribute to an even more pronounced approach to the topic of the special volume (Advanced Materials and Polymers for Industrial Applications).

Due to the insert of new experimental results, the group of authors was increased by one member who contributed to the completion of the research highlighted in the manuscript.

We have added to the manuscript the following information:

In page 4, line 133 to 140 the authors introduced:

 ”Electron microscopy analyses (SEM) used samples of the GFRP composite on the plate directions of the experimental material and another sample of the base material. The samples were coated with a 5 nm Au layer using the Luxor Au SEM COATER - CT-2201-0144 (Kusterdingen, Germany), in order to increase the electrical conductivity of the samples. Surface morphological determinations were performed with the SEM FEI Quanta 200 3D microscope (Brno, Czech Republic) using the following parameters: Low Vaccum mode, LFD (Large Field Detector), HV (high voltage): 20 kV, WD (working distance): 10 mm, Spot size: 5.”

In page 8-9, line 234 to 270 the authors introduced:

 ”Figures 11-12 represent the SEM surface and cross-section images of the GFRP base material, as well as the tensile specimens for the three directions samples. Figure 11a highlights the morphological aspect of the surface of the base material, having a compact, homogeneous surface, without microcracks, but with small porosities. Figure 11b the 10 polymer layers in the GFRP cross-section, which is part of the chemical composition. These microstructures are in agreement with the studies performed by [31].

Figure 11. SEM images of the base material - GFRP: (a) surface image and (b) cross-section image.

Figure 12 shows the cross-section images for the three directions of stress. Thus, Figure 12a highlights the fiber breakage in the longitudinal direction and Figure 12b shows the fiber breakage at 45ᵒ, these 2 breaks having a primary damage aspect. In the case of figure 12c, delamination at the intermediate layers are observed and the breakage is having a medium behavior.

Figure 12. Cross-section SEM images of GFRP specimens in the 3 stress directions: (a) transversal, (b) oriented at 45ᵒ and (c) longitudinal.

Figure 13. Surface SEM images of transversal 0-90 GFRP sample: (a) 100X, (b) 200X, (c) 500X and (d) 2000X

Figure 14. Surface SEM images of oriented at 45ᵒGFRP sample: (a) 100X, (b) 200X, (c) 500X and (d) 2000X

Figure 15. Surface SEM images of longitudinal 0-90 GFRP sample: (a) 100X, (b) 200X, (c) 500X and (d) 2000X

Figures 13-15 illustrate the surface appearance of the tested specimens for all three directions of stress. In all three cases, a similar breakage mechanism is observed in the fi-ber-resin matrix, showing a brittle-type breakage. The different image magnitudes show the fracture behavior of the polymer pull-out fibers at the resin interface and their appearance after breakage. A more compact and evenly distributed morphology of these fibers is observed in figure 13, while figures 14 and 15 show a more pronounced dislocation.”

We hope that we have answered your request, in order to highlight the results to the Journals topics and if you consider something else necessary please don’t hesitate to send us more recomandations.

Reviewer 3 Report

In this paper, the authors have performed experimental testing on the torsion characteristics of a lab-scale turbine blade. Additional numerical modelling was performed to compare and analyze the results. Though the number of results is quite limited, the methodology is correct and the results valuable, despite the low level of elaboration. Major changes are mandatory before acceptance of the paper. Following, a brief list of concerns to be answered by the authors:

1) Details of the turbine blade are not given: aerodynamic profiles, sections, etc.

2) In page 6, the variables employed to determine the normal and shear stresses must be defined and explained (include a small sketch to ease the comparison).

3) In page 6, lines 202 to 205, the symbol for degree has been wrongly converted to an additional zero.

4) The experimental results are not convincingly compared to the FEA results provided by ANSYS. Please, try to provide direct comparison of your experimental results with the simulated ones.

5) Additionally, more details about the numerical model are required: spatial discretizations, mesh characteristics, boundary conditions, convergence and accuracy thresholds…

6) The paper is not ambitious, so the authors should try to provide more novelty or, at least, highlight the significance in a more convenient way in order to improve the overall impact of the paper.

Author Response

Reviewer 3

Dear reviewer,

Thank you very much for your positive feedback. You ask to clarify more details on the GFRP materials and the novelty of our work, which certainly would give an improvement for our research. So, thank you for these valuable comments. We have tried to answer all aspects that were mentioned.

”In this paper, the authors have performed experimental testing on the torsion characteristics of a lab-scale turbine blade. Additional numerical modeling was performed to compare and analyze the results. Though the number of results is quite limited, the methodology is correct and the results valuable, despite the low level of elaboration. Major changes are mandatory before acceptance of the paper. Following, is a brief list of concerns to be answered by the authors:

1) Details of the turbine blade are not given: aerodynamic profiles, sections, etc.

2) In page 6, the variables employed to determine the normal and shear stresses must be defined and explained (include a small sketch to ease the comparison).

3) In page 6, lines 202 to 205, the symbol for degree has been wrongly converted to an additional zero.

4) The experimental results are not convincingly compared to the FEA results provided by ANSYS. Please, try to provide direct comparison of your experimental results with the simulated ones.

5) Additionally, more details about the numerical model are required: spatial discretizations, mesh characteristics, boundary conditions, convergence and accuracy thresholds…

6) The paper is not ambitious, so the authors should try to provide more novelty or, at least, highlight the significance in a more convenient way in order to improve the overall impact of the paper.”

Responses answers 1

In page 5, lines 157 to 172  the authors introduced:

 “A scalable E-glass composite blade structure and reinforcement system was designed. The dimensions of the wind turbine blade (WTB) have been analyzed and the blade and the rigging (reinforcing system), as well as the attached devices (blade matrix and moldings for tensile and torsional stresses), have been designed. The blade has an overall length of 1750mm, a maximum chord of 273.71mm, chord tip at 81.54mm, an angle of entry of 55.62° and an angle of attack 29.74°; with full pitch control to reduce aerodynamic cyclic loads. Blade attachment to the rotor is via a 100mm diameter coupling. The blade is designed with a right leading edge and a right taper trailing edge. A NACA airfoil profile was used for structural performance. In order to achieve increased structural strength and stiffness at 0.286R (R - total blade radius - distance from the center of the rotor to the tip of the blade), the same NACA profile was applied to both the upper and lower surfaces. The configuration of the wind turbine blade (WTB) type GE1.5sle was determined by studying the parameters using the finite element method (FEM). The FEM analysis proved that the joint is safe under twist loading conditions. The determination of the critical zones of the WTB scalable during load simulation using FEA shows that the constituent elements of the respective airfoil I profile respect the same critical zones.”

Responses answer 2

In page 10, lines 303 to 304  the authors introduced:

The variables employed to determine the normal and shear stresses have been defined and explained on page 10 , using figure 8b (the figure existing in the initial version of the manuscript), “where εa, εb and εc are deformations measured with the three strain gauge marks of the rosette in Figure 8b.”

Responses answer 3

In page 10, line 313 to 315  we changed from 0 to degrees.

Responses answer 4

In page 12, line 329 to 338  the authors introduced:

 “In the finite element analysis (figure 7) it is found that the highest value of the von Mises stress is obtained at the tip of the blade, corresponding to the location of the T4 transducer (figure 10). Considering the von Mises relation: , from the experiment we deduce that:

Formula

where the stresses calculated from the strains provided by the T4 transducer at a torque of 59.38 N·m are approximately: σmax=80 MPa and τmax=60 MPa. It is found that close values of the von Mises stresses from the FEA and the experiment are obtained, respectively:

AEF: Torque: 60 N·m, σvM=139.66 MPa; Test: Torque: 59.38 N-m, σvM=131.14 MPa.”

Responses answers 5

In page 6, line 173 to 186  we introduced:

 “The blade was considered fixed in the hub and a maximum torque (torsion moment) of 60 Nm was applied at its top, Figure 5. The torsion moment is oriented along of the Oz axis, and the direction of rotation was in trigonometric direction (market with 2 at test) and clock-wise direction (market with 1 at test).

Figure 5. Support and loading scheme

From the discretization step a number of 1338842 nodes and 832563 elements were determined. The maximum element size was 8 mm for the turbine blade and the Sizing function was used to increase the number of elements in the required areas: on the trailing edge 0.5 and 1 mm elements and in the transition area from the cylindrical part of the blade to the blade profile 3 mm elements. For the hub in the embedment area and on the ridig structure we used 5mm elements, Figure 6.

Figure 6. Discretization of the model”

Responses answers 6

The novelties introduced in the paper were in chapters 2 and 3: the material from which the blade and the specimens were made were presented.  Electron microscopy analyses (SEM) for morphological aspects of the tested samples and tensile tests were carried out on materials made of fiberglass in order to see the internal structure of the material.

In page 3-4, line 113 to 130 the authors introduced:

 ”The material used in the manufacture of the blade and specimens is RT 500 glass fiber reinforced composite (GFRP) and EPIKOTE epoxy resin. Three test specimens were cut from the composite plate, in three directions (transverse - TR, longitudinal - LG and at 45°), Figure 2 a, composed of 10 layers oriented at [0°/90°], Figure 2b.

(a)

(b)

Figure 2. (a) Directions for sampling from the GFRP plate; (b).and configuration of layers composite [0 ° / 90 °].

Tensile Tests

Three samples were tensile tested, with specimen sizes conforming to ASTM D3039. The tests were carried out on the INSTRON 8801 universal machine that developing a maximum force of 100KN. The test regime was set by increasing the extension rate to 0.5mm/min. Due to the constant speed of movement during the test we have the possibility to observe the occurring deformations and a good accuracy of the drawing of the characteristic curve. The stresses were held until breakage, as can be seen in Figure 3.”

Figure 3. Typical failure modes of GFRP laminates: (a) transversal; (b) longitudinal. and (c) oriented at 45°.

In page 4, line 133 to 140 the authors introduced:

 ”Electron microscopy analyses (SEM) used samples of the GFRP composite on the plate directions of the experimental material and another sample of the base material. The samples were coated with a 5 nm Au layer using the Luxor Au SEM COATER - CT-2201-0144 (Kusterdingen, Germany), in order to increase the electrical conductivity of the samples. Surface morphological determinations were performed with the SEM FEI Quanta 200 3D microscope (Brno, Czech Republic) using the following parameters: Low Vaccum mode, LFD (Large Field Detector), HV (high voltage): 20 kV, WD (working dis-tance): 10 mm, Spot size: 5.”

In page 8-9, line 234 to 270 the authors introduced:

 ”Figures 11-12 represent the SEM surface and cross-section images of the GFRP base material, as well as the tensile specimens for the three directions samples. Figure 11a highlights the morphological aspect of the surface of the base material, having a compact, homogeneous surface, without microcracks, but with small porosities. Figure 11b the 10 polymer layers in the GFRP cross-section, which is part of the chemical composition. These microstructures are in agreement with the studies performed by [31].

Figure 11. SEM images of the base material - GFRP: (a) surface image and (b) cross-section image.

Figure 12 shows the cross-section images for the three directions of stress. Thus, Figure 12a highlights the fiber breakage in the longitudinal direction and Figure 12b shows the fiber breakage at 45ᵒ, these 2 breaks having a primary damage aspect. In the case of figure 12c, delamination at the intermediate layers are observed and the breakage is having a medium behavior.

Figure 12. Cross-section SEM images of GFRP specimens in the 3 stress directions: (a) transversal, (b) oriented at 45ᵒ and (c) longitudinal.

Figure 13. Surface SEM images of transversal 0-90 GFRP sample: (a) 100X, (b) 200X, (c) 500X and (d) 2000X

Figure 14. Surface SEM images of oriented at 45ᵒGFRP sample: (a) 100X, (b) 200X, (c) 500X and (d) 2000X

Figure 15. Surface SEM images of longitudinal 0-90 GFRP sample: (a) 100X, (b) 200X, (c) 500X and (d) 2000X

Figures 13-15 illustrate the surface appearance of the tested specimens for all three directions of stress. In all three cases, a similar breakage mechanism is observed in the fi-ber-resin matrix, showing a brittle-type breakage. The different image magnitudes show the fracture behavior of the polymer pull-out fibers at the resin interface and their appear-ance after breakage. A more compact and evenly distributed morphology of these fibers is observed in figure 13, while figures 14 and 15 show a more pronounced dislocation.”

Due to the insert of new experimental results, the group of authors was increased by one member who contributed to the completion of the research highlighted in the manuscript.

We hope that we have answered all the topics you mentioned and also, we have added to the paper some material analysis in order to highlight the results to the Journals topics.

Round 2

Reviewer 2 Report

The great effort and commitment of the authors in the general improvement of the paper is commendable and remarkable, not only by correcting significant scientific errors detected by the reviewers, but also by including additional results (SEM results) in the paper. It would be beneficial to clarify the type of element used in the FEA analyses and whether or not the numerical results resulted from a previous mesh sensitivity analysis. Please make the legend of figure 7 legible. Does this legend refers to stresses or displacements values?

Author Response

Response to Reviwer 2

Dear reviewer,

We bring to you these second issues that you want to clarify:

  1. It would be beneficial to clarify the type of element used in the FEA analyses and whether or not the numerical results resulted from a previous mesh sensitivity analysis.

Response 1.

In page 6, line 179 to 181, the authors introduced  “Elements in FEA are generally grouped into 1D elements, 2D elements and 3D elements. They are recognised according to their shapes. In the present case we used 3D solid tetrahedral elements”.

There have been some previous analyses where convergence of the solution in relation to the number of elements has been observed. In any case, the number of elements used (832563) is large enough to obtain a more accurate value of the results, here the von Mises stress.

  1. Please make the legend of figure 7 legible. Does this legend refers to stresses or displacements values?

Response 2.

We have corrected figure 7 to make it, now, legible. It can now be seen that the legend refers to von Mises stresses.

We hope that we have answered your second request, in order to give the final the final aspect and to meet the Journals topics.

Reviewer 3 Report

The authors have addressed the major concerns reasonably so now the paper can be accepted for publication.

Author Response

Dear reviewer,

Thank you very much for your observations. We think that the paper will add value to the topics of Polymers Journal in terms of FEA analysis and GFRP materials.

Best regards,